# The effect of ascertainment on penetrance estimates for rare variants: Implications for establishing pathogenicity and for genetic counselling

**Andrew D. Paterson**[1,2]*, **Sang-Cheol Seok**[3], **Veronica J. Vieland**[3,4]

**1** Program in Genetics and Genome Biology, The Hospital for Sick Children, Toronto, Ontario, Canada,
**2** Divisions of Epidemiology and Biostatistics, Dalla Lana School of Public Health, University of Toronto, Toronto, Ontario, Canada, **3** Mathematical Medicine LLC, Chicago, IL, United States of America,
**4** Departments of Pediatrics and Biostatistics (Emerita), The Ohio State University, Columbus, OH, United States of America

* andrew.paterson@sickkids.ca

**Data Availability Statement:** Data are simulated. Code used in generating and analyzing the data is available at https://github.com/MathematicalMedicine/PenetranceEstimator/.

## Abstract

Next-generation sequencing has led to an explosion of genetic findings for many rare diseases. However, most of the variants identified are very rare and were also identified in small pedigrees, which creates challenges in terms of penetrance estimation and translation into genetic counselling in the setting of cascade testing. We use simulations to show that for a rare (dominant) disorder where a variant is identified in a small number of small pedigrees, the penetrance estimate can both have large uncertainty and be drastically inflated, due to underlying ascertainment bias. We have developed PenEst, an app that allows users to investigate the phenomenon across ranges of parameter settings. We also illustrate robust ascertainment corrections via the LOD (logarithm of the odds) score, and recommend a LOD-based approach to assessing pathogenicity of rare variants in the presence of reduced penetrance.

## Introduction

Next-generation sequencing has led to an explosion in the number of genetic findings for many rare diseases. For certain types of rare coding variants (e.g. missense, or protein truncating), if the variant is sufficiently rare and has bioinformatic predictions that are severe, current algorithms result in it being classified as pathogenic [1]. However, the analysis of large-scale sequencing from cohorts, such as ExAC [2], gnomAD [3], and the UK Biobank [4], has shown that many such variants may often lack clinically significant impact. For example, ExAC estimated that individuals from population cohorts carried a mean of 53 variants previously thought to be sufficient causes of Mendelian diseases [2]. Additionally, 88% of such variants had MAF>1%, implying that they are likely not sufficient causes. This may indicate that such variants are not causally related to disease, or perhaps, that they are causally related but with reduced penetrance.

Penetrance plays an important role in understanding disease pathology, in the appropriate classification of pathogenic variants, and perhaps above all in the context of genetic counseling.

**Funding:** Funding came from the National Institutes of Health (NINDS NS085238) in the form of salary support for VJV and SCS. Salary support to VJV and SCS came via a subcontract to Mathematical Medicine from the NIH grant listed above. The funders had no role in study design, data collection and analysis, decision to publish, or preparation of the manuscript. The specific roles of these authors are articulated in the 'author contributions' section.

**Competing interests:** Salary support to VJV and SCS came via a subcontract to Mathematical Medicine from the NIH grant (NS085238). This does not alter our adherence to PLOS ONE policies on sharing data and materials. There are no patents, products in development or marketed products associated with this research to declare.

However, most of the variants reported to date have been very rare and identified in small sets of unrelated individuals (sometimes just one) or small pedigrees. Penetrance cannot be estimated from a single case, or a single parent-offspring trio presenting with a *de novo* mutation in the offspring. But even with multiple cases or families, determination of the penetrance can present challenges. Here we focus on one such challenge: ascertainment.

Typically a variant of interest is first identified in one individual with a given phenotype. Investigators may then sequence either additional relatives of the individual, or additional individuals or families presenting with the same or closely related phenotypes, with the goal of bolstering the case for pathogenicity. Thus, ascertainment of individuals to be sequenced typically proceeds in stages. The precise ascertainment process used to enroll individuals and/or families is usually at least to some extent unsystematic, and may vary between families. Ascertainment is therefore challenging to model when attempting to estimate the penetrance of a variant.

One situation in which ascertainment can be easily handled is "single" ascertainment, in which the probability of an affected individual being ascertained is proportional to the number of affected individuals in the family [5]. In fact, much of the literature on inferring pathogenicity or estimating penetrance tends to assume single ascertainment, e.g., [6], where ascertainment is addressed by conditioning on "the proband," a procedure which is strictly correct only under true single ascertainment. While it is true that the typical study ascertains families through one individual who may be designated as the single "proband", this does not ensure that the study meets the proportionality requirement of single ascertainment. This requirement would be violated, e.g., if families with four affected members were more than twice as likely to be recruited as families with just two; or, if the probability of a second sibling being ascertained were dependent on the ascertainment status of the first. And in general, if either (i) ascertainment is not truly single, or (ii) even if it is, if an appropriate ascertainment correction is not incorporated into the estimation method, then penetrance estimates will be biased. Here we consider the magnitude of that bias, across a range of plausible ascertainment models and varying amounts of available data.

Penetrance estimation also plays a role in the assessment of pathogenicity. Some approaches to the interpretation of rare coding variants assume either full or high penetrance [7], for the sake of simplicity. Extensive criteria have been proposed to claim a causal relationship between variants and disease, and the authors have urged caution in presuming full penetrance for pathogenic variants [8]. But in practice, penetrance remains an important consideration. For instance, the ACMGG/AMP joint consensus recommendations [1] warns against ignoring the possibility of reduced penetrance in establishing segregation of a VOI with a phenotype, but also instructs that "lack of segregation...provides strong evidence against pathogenicity." (p. 15). And in practice, many laboratories will rule out candidate VOIs when they are found among unaffected relatives. Particularly in the absence of a rigorous and accurate estimate of the actual penetrance, this complicates the use of segregation information in assessments of pathogenicity. Below we consider some implications for the assessment of pathogenicity in the presence of reduced penetrance, and we propose a new metric for assessing co-segregation between a VOI and disease.

## Methods

### Preliminaries and notation

We focus here on sibship data. The impact of ascertainment for more complex pedigrees can be approximated by considering large sibship sizes. We assume a very rare variant of interest (VOI), and an autosomal dominant disease D. Let a qualifying individual (QI) be anyone who

is both heterozygous (HET) for the VOI and also affected (AFF) with D. Let $r$ be the number of QI sibs within a family, and let $t$ be the number of AFF sibs regardless of VOI genotype. We also assume that, regardless of VOI status, an individual might develop D due to other factors, which might be genetic (involving one or more VOIs at other loci or other variants within the same gene) and/or environmental (e.g., due to infections). Let $\gamma$ be the combined penetrance across all causes other than the VOI under study. Since we assume the VOI is very rare, $\gamma$ is effectively the population prevalence of D. Let $s$ be the total number of siblings in a family (regardless of phenotype), and let N be the number of $s$-sized sibships in a dataset.

## Ascertainment model

In order to consider a range of plausible ascertainment scenarios, we employ the general family-based $k$-model of ascertainment [9]. In its simplest form, this model stipulates that the probability that a family is ascertained is proportional to $r^k$, where $k$ controls the model. For example, when $k = 1$, the probability of ascertainment is strictly proportional to $r$: this is equivalent to classical "single ascertainment". Similarly, when $k = 0$, so that every family with $r \geq 1$ is ascertained, this model is equivalent to classical "complete" or "truncate" ascertainment. We generalize this model in two ways. First, we assume that ascertainment requires $r \geq 1$, that is, every ascertained family contains at least one QI, but we allow that there may be additional preferential ascertainment of families based on $t$ alone, that is, that investigators may preferentially ascertain families with more affected individuals without knowing (or prior to knowing) the VOI status of those additional individuals. Second, we allow that even an individual carrying the VOI may develop disease due to any other independent causes at work in the general population. With these two extensions in mind, our ascertainment model becomes

P[sibship is ascertained | $r$, $t$] = $c(r^k+t)$; for $r \geq 1$, and 0 otherwise where $c$ is a normalizing constant.

## Estimation methods

Let $f$ be the attributable penetrance, or the penetrance due to the VOI for HET individuals. (Note that when $\gamma > 0$, $\beta$ = P[AFF|HET] = $\gamma + f - \gamma f$. However, we focus here on estimation of $f$ itself rather than $\beta$.) In what follows, we estimate $f$ in three ways, the first two of which are:

(i) $\tilde{f}$ is obtained by counting the proportion of AFF individuals among all HET individuals in the data set, after dropping one QI individual per family, that is, applying the correction for single ascertainment;

(ii) $\tilde{f}^*$ is obtained by counting the proportion of AFF individuals among all HET individuals in the data set, that is, without applying any ascertainment correction.

$\tilde{f}^*$ is a naïve estimate, which would be correct if the families were not ascertained based on either phenotype or genotype. It is clearly, however, incorrect under any of our ascertainment models. Our interest in this estimate is to establish how biased it becomes under various ascertainment scenarios. $\tilde{f}$ by contrast, does apply the frequently employed single ascertainment correction, and again, our interest in $\tilde{f}$ is to establish how biased it will be under ascertainment scenarios other than single ascertainment.

The third form of penetrance estimate we consider is based on an "ascertainment assumption free" [10] approach, which involves conditioning on all of the phenotypic data. This is the ascertainment correction implicit in the usual LOD score [11–13], and also the LOD score allowing for linkage disequilibrium or LD-LOD [6, 14, 15], and in principle any program that allows calculation of the LOD score will support this method. The calculation is done here

assigning the VOI (which plays the role of the "marker") and the disease allele the same (rare) frequency (we have used 0.001 in the simulations), assuming complete linkage disequilibrium between the two (D′ = 1), and also assuming 0 recombination between the marker and the disease allele. Free parameters in the model are then the 3 penetrances; in our calculations we also include the admixture parameter $\alpha$ of Smith [16], representing the probability that any given family is of the "linked" type, which adds robustness when phenocopy levels are high.

Maximizing the LD-LOD over the free parameters gives us the LD-MOD, which occurs at the maximum likelihood estimate (m.l.e.) of $\hat{f}$ of $f$ [10–13], giving us our third estimate:

(iii) $\hat{f}$ is obtained by maximizing the LD-LOD over the penetrance vector.

## Assessment of pathogenicity

While maximizing the LD-LOD can be used to estimate $f$, the LD-MOD itself is not a good statistic for representing the strength of evidence for co-segregation between the VOI and disease, because it is not additionally conditioned on ascertainment through the VOI. We note, however, that in nuclear families, once we ascertain so as to require the VOI to be present in the family, there is no remaining LD information in the sibship, since LD information is conveyed entirely by the marker allele frequencies in the parents. Therefore, assessments of co-segregation can be made using the ordinary (linkage equilibrium) LOD, or LE-LOD. Because maximizing the LE-LOD itself will not return true m.l.e.s of $f$ under the LD model, we consider evaluating the LE-LOD at the maximizing model obtained from the LD-MOD, for a statistic we annotate as LE-LOD(max).

Thompson et al. [6], following Petersen et al. [15], proposed using a particular form of what they refer to as a Bayes Factor (BF) for assessing the strength of evidence for co-segregation of the VOI with disease. We refer to this statistic as the Thompson BF (TBF). The TBF is closely related to the LD-LOD, but it incorporates an additional adjustment for single ascertainment through a QI. As we illustrate below, unlike the LD-LOD, the TBF cannot be maximized to obtain ascertainment-corrected estimates of the penetrances; and [6] did not recommend using it for this purpose. However, this complicates application of the TBF, which requires specifying a fixed set of pentrances (but see also [15]), which must be separately obtained or estimated; furthermore, it is not clear whether the adjustment for single ascertainment incorporated into the TBF is strictly correct or robust to other ascertainment models.

In what follows we evaluate the behavior of the LE-LOD(max), and compare it with the TBF, using the simulated data. For comparability with the LOD, we report TBF on the $\log_{10}$ scale. We also incorporate the admixture parameter $\alpha$ in order to afford comparability with the LE- or LD-LOD in maintaining some degree of robustness to phenocopies.

## Simulation methods

Expected values of $\tilde{f}, \tilde{f}^*$ and $\hat{f}$ were obtained via simulation, by averaging each estimate's value across 1,000 replicates per generating condition, and standard errors were obtained by averaging the standard deviation of each estimate across those same 1,000 replicates. (While the expected values of $\tilde{f}$ and $\tilde{f}^*$ are easily calculated analytically, the standard errors are not.) We note that, depending on the generating conditions, many sibships may end up with only the QI being HET. In this case, the proportion of AFF out of all HET individuals cannot be scored after dropping the QI, and therefore any such sibships do not contribute to $\tilde{f}$ and $\tilde{f}^*$, effectively reducing the sample size. For simplicity, in computing $\tilde{f}$ and $\tilde{f}^*$ we assume all parents are

phenotypically and genotypically unknown; when computing LODs and TBF, parents are treated as genotypically known but phenotypically unknown. Including parental information does not substantively affect results. Simulations and calculations were done in MATLAB (2021.9.10.0.1739362 (R2021a), Natick, Massachusetts: The MathWorks Inc.); LE-LOD and TBF calculations were done using KELVIN [17].

## Results

### Impact of ascertainment on penetrance estimates

Fig 1 shows results for true single ascertainment (k = 1), for s = 2, as a function of sample size N. Here we assume that the true value of $f$ = 0.5. As can be seen, in this case, the mean of $\tilde{f}$ = 0.5, the generating value, as expected. But using $\tilde{f}^*$ the estimates are seriously upwardly biased in all data sets, regardless of N. Note that because each sibship contains at least one QI, by stipulation, the minimum value of $\tilde{f}^*$ is 0.50.

Note too that even the correct estimate $\tilde{f}$ shows considerable sampling variability. For instance, with N = 10, $\tilde{f}$ will be >70% or <30% in approximately 40% of all data sets when $f$ = 50%. This variability remains appreciable even for N = 50.

For ascertainment models other than single, overall variability remains similar to what is shown in Fig 1, but even $\tilde{f}$ tends to be biased, with mean $\tilde{f}$ = 0.60, 0.50, 0.43 and 0.38 for k = 2, 1, 0 and −1, respectively. In all cases, the uncorrected $\tilde{f}^*$ will return even more biased estimates, with mean $\tilde{f}^*$ = 0.89, 0.88, 0.87 and 0.86, for k = 2, 1, 0 and −1, respectively.

Fig 2 shows the impact of the population prevalence $\gamma$ on average penetrance estimates. Focusing first on single ascertainment (k = 1) and $f$ = 0.5, we can see that regardless of k, the

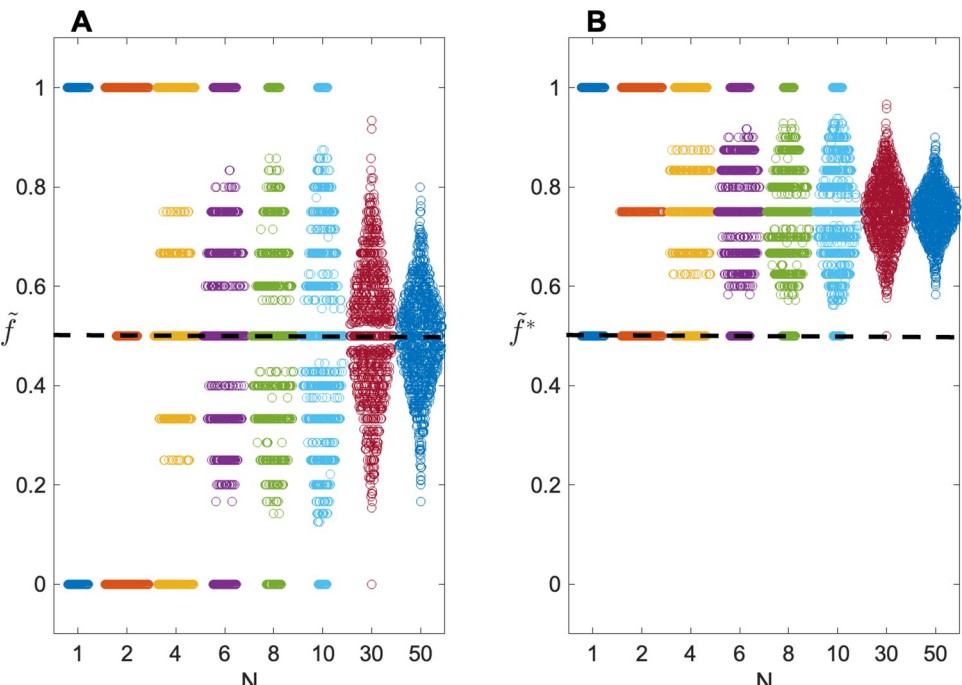

**Fig 1. Swarm plots showing sampling distributions of penetrance estimates as a function of number of families N.**
Distributions of (A) $\tilde{f}$ and (B) $\tilde{f}^*$ are shown for simulations of 1000 replicates, with true penetrance $f$ = 0.5. The number of sibs per family s = 2; phenocopy rate $\gamma$ = 0. Users interested in varying the parameters can use the PenEst app.

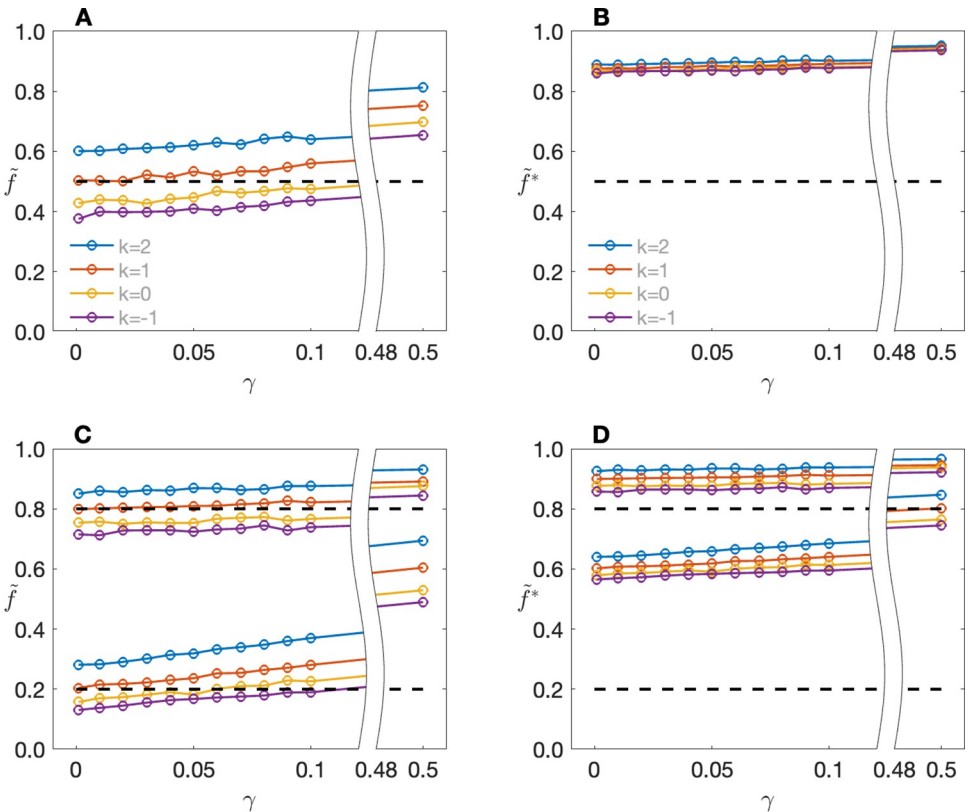

**Fig 2. Expected values of penetrance estimates as a function of population prevalence $\gamma$ and ascertainment parameter $k$.** Expected values of (A) $\tilde{f}$ and (B) $\tilde{f}^*$ when the true penetrance $f = 0.5$; expected values of (C) $\tilde{f}$ and (D) $\tilde{f}^*$ when $f = 0.2$ (lower line sets) or $f = 0.8$ (upper line sets). The number of sibs per family, $s = 2$. Users interested in varying the parameters can use the PenEst app.

expected value of $\tilde{f}$ is relatively independent of $\gamma$ until $\gamma$ becomes quite high. Note that for $f = 0.5$ and $\gamma = 0.5$, the actual probability that a VOI carrier is affected under our generating model is $0.5 + 0.5 - (0.5)(0.5) = 0.75$, which is in line with the estimates returned by $\tilde{f}$. $\tilde{f}^*$ might be said to be even more robust to $\gamma$, although this is because in this case $\tilde{f}^*$ is already close to the top of the scale for $\gamma = 0$. Moreover, $\tilde{f}^*$ appears not only indifferent to $\gamma$, but also to $f$ itself, with estimates >70% even for $f = 0.05$, and >80% for $f = 0.05$ when $\gamma = 0.5$. These patterns repeat for different values of $k$, with visible impact only on the magnitude of the bias for any given $(f, \gamma)$ combination. Ascertainment effects will be reduced as $s$ increases. Users who are interested in investigating penetrance estimates for other ascertainment models, other combinations of parameter values or other sibship sizes are encouraged to download the PenEst app: https://github.com/MathematicalMedicine/PenetranceEstimator.

Fig 3 shows results for $\hat{f}$ for the same data used in Figs 1 and 2. As can be seen, $\hat{f}$ behaves very much like $\tilde{f}$ when k = 1 (Fig 3A), but it retains almost complete robustness to ascertainment, and also to $\gamma$ at least until $\gamma$ is quite large (Fig 3B). (As with $\tilde{f}$, as $\gamma$ gets very large, $\hat{f}$ covers both cases due to the VOI and those among variant carriers due to other causes.) Comparing Fig 3A with Fig 1A, $\hat{f}$ shows slightly greater sampling variability than $\tilde{f}$; this is due to the inherent ascertainment correction built in to $\hat{f}$. The slight but systematic over- or under-estimation of $f$ seen in Fig 3B is due to the small sample size; as N increases $\hat{f} \rightarrow f$

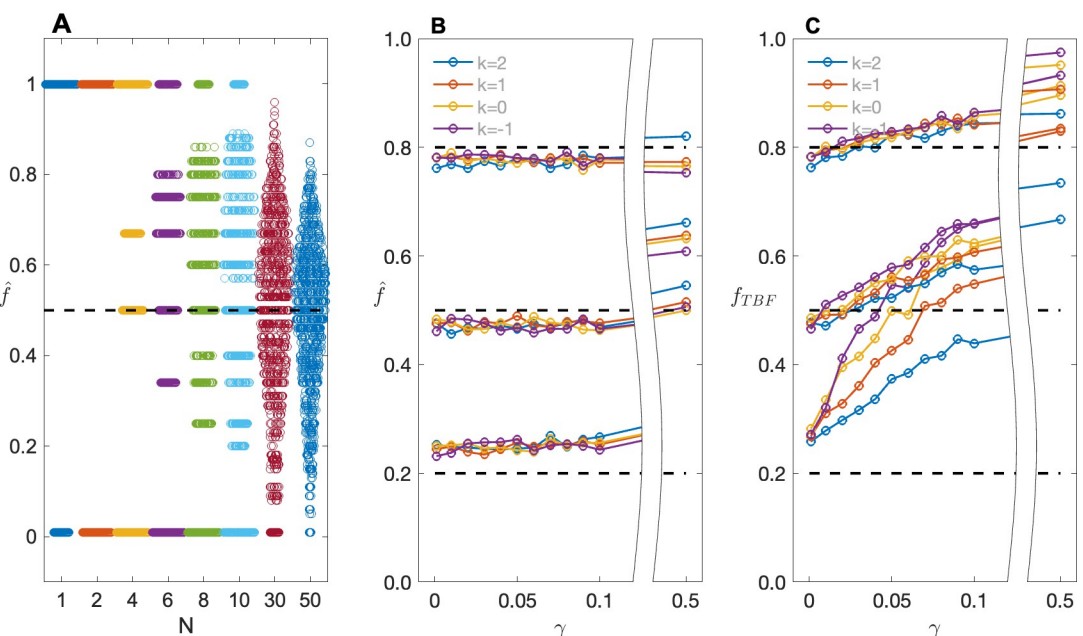

**Fig 3. Sampling distributions and expected values of $\hat{f}$.** (A) Swarm plots showing sampling distributions of $\hat{f}$, as obtained from maximizing the LD-LOD, as a function of number of families N; (B) Expected values of $\hat{f}$ as a function of population prevalence $\gamma$ and ascertainment parameter $k$, for $f$ = 0.2, 0.5 and 0.8, reading from bottom to top of the plot, respectively; (C) Expected values of the estimate of $f$ obtained by maximizing the TBF (denoted here as $f_{TBF}$) as a function of population prevalence $\gamma$ and ascertainment parameter $k$, for $f$ = 0.2, 0.5 and 0.8, reading from bottom to top of the plot, respectively. Data are the same as used to generate Figs 1 and 2, respectively.

(results not shown). However, in small samples the upward bias can be appreciable particularly when $f$ is small; e.g., when $f$ = 0.05 ($\gamma$ = 0), for N = 20, the expected value of $\hat{f}$ = 0.165.

For comparison purposes, Fig 3C shows the corresponding results based on maximizing the TBF. As noted above, this procedure has never been proposed as a mechanism for estimating $f$, but the figure illustrates that the small differences in form between the LD-LOD and the TBF fundamentally change the applicability of the ascertainment assumption free approach to estimation. This becomes relevant when deciding how to set parameter values in calculating the TBF for purposes of assessing pathogenicity. We note too that, particularly in the presence of phenocopies, estimates obtained by maximizing the TBF remain highly biased even in very large samples. For example, for s = 2, k = 1, N = 1000 and $f_{DD} = f_{Dd}$ = 0.05 or 0.5, when $\gamma$ = 0.1, maximizing the LD-LOD returns estimates of $f_{Dd}$ of 0.07 (s.d. 0.05) and 0.50 (0.04), respectively, while maximizing the TBF returns 0.74 (0.26) and 0.66 (0.12), respectively.

## Assessment of pathogenicity

Fig 4A shows the distribution of the LE-MOD(max) as a function of $\gamma$ and $k$, for $f$ = 0.5, s = 2 and N = 20. Not surprisingly, as $\gamma$ increases, evidence for co-segregation decreases; also notable is that, while estimates of $f$ are robust to ascertainment, the LE-LOD(max) itself increases as $k$ increases; but since there really is co-segregation, this is not in itself problematic. While values of LE-MOD(max) are small (see also below), they are consistently positive until $\gamma$ is quite large, indicating evidence in favor of co-segregation. By contrast, results for the TBF(gen) (Fig 4B) become increasingly negative as $\gamma$ increases, erroneously indicating evidence against co-segregation for even small values of $\gamma$, with strikingly negative values for large $\gamma$. For

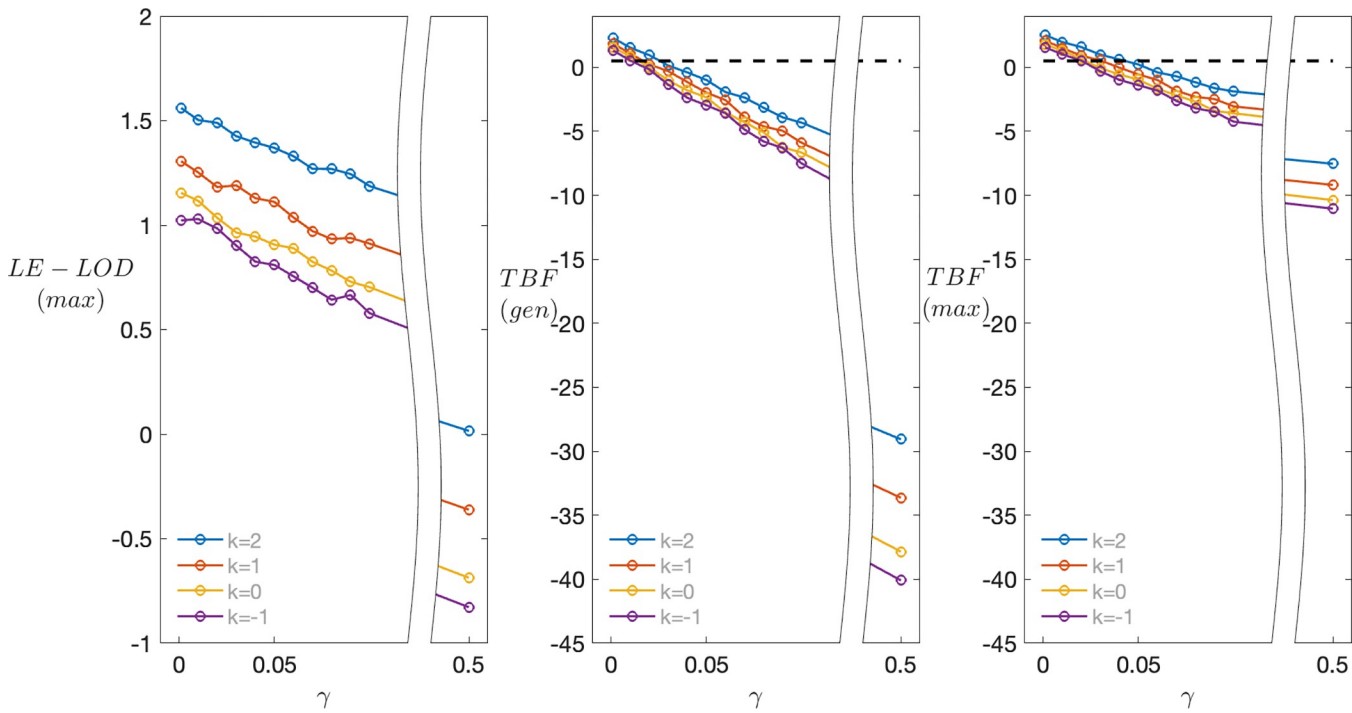

**Fig 4. Expected values of alternative co-segregation measures as a function of ascertainment model and phenocopy rate for N = 20.** (A) Expected values of LE-LOD(max) as a function of population prevalence $\gamma$ and ascertainment parameter $k$, for $f = 0.5$, $s = 2$ and N = 20. (B) Expected values of the TBF(gen) for the same data. (C) Expected values of the TBF evaluated at the same maximizing model used to evaluate LE-LOD in panel A. Note the different scales on the y-axis across subplots.

comparison we also show (Fig 4C) results for the TBF when it is evaluated at the same maximizing model used to calculate LE-LOD(max). While this ameliorates the problem somewhat, especially for large $\gamma$, the basic pattern of results remains the same.

Fig 5 shows the distribution of LE-LOD(max), as a function of N, when data are generated under the alternative hypothesis of (complete) disequilibrium (Fig 5A) and the null hypothesis of no linkage and no disequilibrium (Fig 5B), for $s = 2$, $k = 1$, $\gamma = 0$, and $f = .5$. Notably, under the alternative hypothesis, evidence of co-segregation of the VOI with disease tends to be quite weak until N is at least 30, and even then the chance of obtaining a small LOD score remains high. Under these generating conditions, it apparently requires closer to 50 2-child families before there is a reasonable chance of obtaining a substantial LE-LOD(max). Under the null distribution, even with N = 50 LE-LOD(max) scores are not consistently negative. However, the maximum and minimum scores all remain small in magnitude, so that the distributions under the alternative and the null are increasingly non-overlapping. For example, when N = 50 and there is co-segregation, 480/1000 replicates return LE-LOD(max) $\geq 3$; however, when there is no co-segregation, 0 out of 1000 replicates do so.

## Discussion

In general, our simulations show that under unsystematic ascertainment schemes, or in cases where appropriate ascertainment corrections are not included in the estimation procedure, there is a high risk of over-estimating the penetrance of any given VOI. This finding is consonant with, and may in large part explain, reports for specific variants. For example, multiple coding variants in *PRNP* had been reported to cause rare dominant monogenic

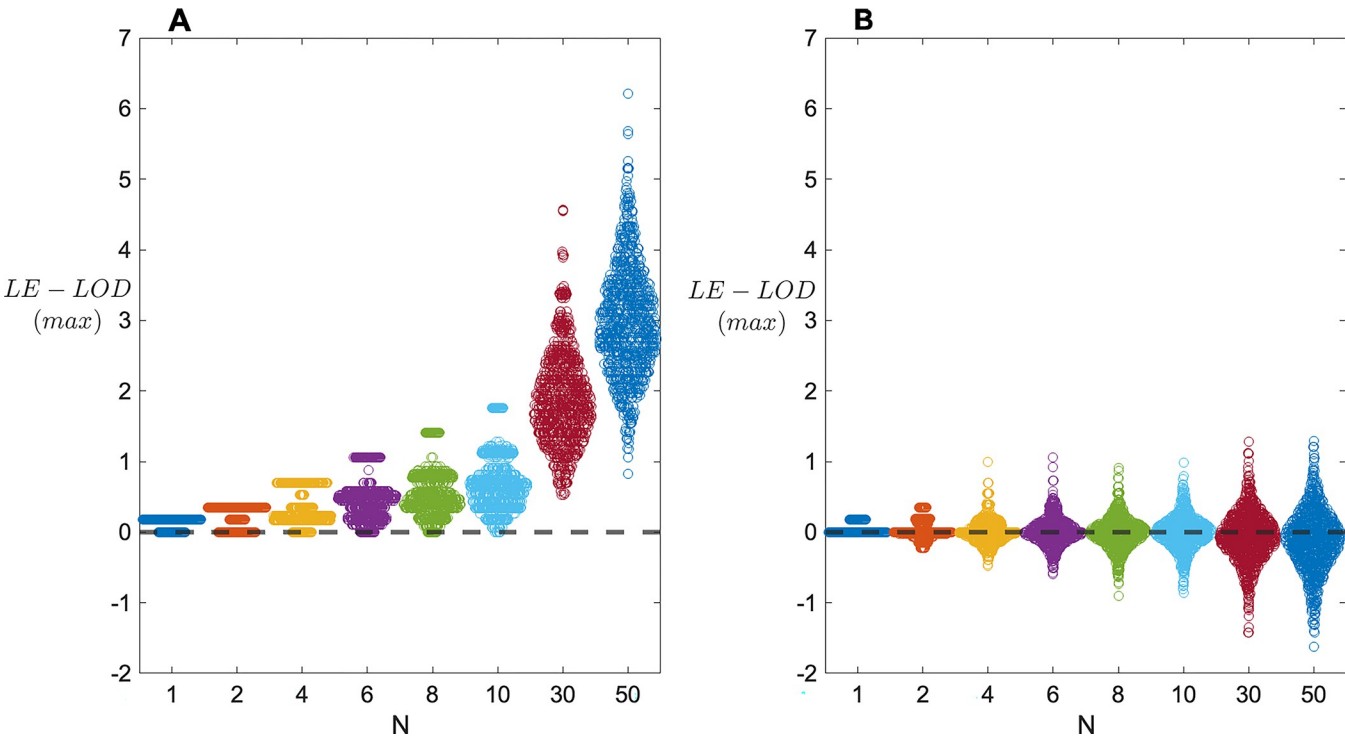

**Fig 5. Sampling distributions of LE-LOD(max) as a function of number of families N.** (A) Results when data are generated under the hypothesis of (complete) disequilibrium, with s = 2, k = 1, γ = 0, and f = .5. (B) Corresponding results when data are generated under the null hypothesis of no linkage and no linkage disequilibrium.

neurodegenerative disease, but there was a 30-fold higher prevalence of variants previously suggested to be causal in this gene in ExAC compared to the expected frequency calculated from the estimated prevalence of the disorder [18]. Specifically for three variants the lifetime risk of developing disease was <10%. Similarly, GWAS array data from the UK Biobank were used to estimate pathogenicity, penetrance, and expressivity of putative disease-causing rare variants (MAF<1%) that were directly genotyped and had good quality [19]. Focused on maturity-onset diabetes of the young and developmental disorders, many specific variants were found for which the penetrance—estimated either in families ascertained for the presence of the VOI or in disease cohorts—was much higher than that obtained from a population-based cohort. For example, previous studies had estimated the penetrance of *HNF4A* rs137853336 (chr20:43042354C>T, p.Arg114Trp) to be up to 75% by age 40 years from a large Maturity Onset Diabetes of the Young cohort, but data from the UK Biobank estimated penetrance to be <10% [19]. Similarly, in the same study, none of 6 protein truncating variants in 5 genes that had previously been related to disease via a haploinsufficiency mechanisms were associated with development traits, casting doubt that such variants in these genes are a cause of developmental delay.

In another study, the median penetrance was estimated to be 14% for 361 variants that were observed in multiple individuals from genes in which some variants are related to either hypertrophic or dilated cardiomyopathy [20]. For example, *MYBPC3*:c.1504C>T:p.R502W, had penetrance estimated of ~50% by age 45 years in the clinical setting. However, penetrance estimates of 6.4% were obtained for this variant from two population-based sequencing cohorts. The extent of coding variation in humans is astounding: gnomAD shows that on average each individual harbours around 11,000 missense variants, about 200 of which are rare (allele

frequency <0.1%) [21]. Unique variants are also relatively common: each participant in gnomAD has a mean of 27 (±13) novel coding variants that were not observed in other individuals in gnomAD [21]. These observations have implications for genetic counselling, including the recommendation of invasive screening procedures and administration of preventative treatment.

By contrast, maximizing the LD-LOD over the penetrances, which yields the LD-MOD, is a valid method for obtaining ascertainment-adjusted maximum likelihood estimates. Variability of these estimates remains a concern, however, even in reasonably large sample sizes (say, N = 50 sibships). While the LD-MOD itself cannot be used as a measure of evidence for or against co-segregation, because it is not properly conditioned on ascertainment through the VOI, the penetrance estimates obtained from the LD-MOD can be used in conjunction with the ordinary (linage equilibrium) LOD to give a statistic we called the LE-LOD(max). This statistic appears to perform more reliably than the Bayes factor proposed by Thompson et al. [6] in application to sibship data under the conditions we have simulated in this paper. It reminds us, however, that in the presence of reduced penetrance, attributions of co-segregation between a VOI and a disease can be difficult to reliably establish, or rule out, without substantial quantities of data.

## Acknowledgments

Special thanks to Jo Valentine-Cooper for creation of the PenEst app.

## Author Contributions

**Conceptualization:** Andrew D. Paterson, Veronica J. Vieland.

**Formal analysis:** Sang-Cheol Seok, Veronica J. Vieland.

**Investigation:** Andrew D. Paterson, Sang-Cheol Seok, Veronica J. Vieland.

**Methodology:** Sang-Cheol Seok, Veronica J. Vieland.

**Software:** Sang-Cheol Seok, Veronica J. Vieland.

**Supervision:** Andrew D. Paterson, Veronica J. Vieland.

**Visualization:** Sang-Cheol Seok.

**Writing – original draft:** Andrew D. Paterson, Veronica J. Vieland.

**Writing – review & editing:** Andrew D. Paterson, Sang-Cheol Seok, Veronica J. Vieland.

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
