## [Decision Letter · Decision Letter 0]

14 Jun 2023

PONE-D-23-06786The effect of ascertainment on penetrance estimates for rare variants: implications for establishing pathogenicity and for genetic counsellingPLOS ONE

Dear Dr. Paterson,

Thank you for submitting your manuscript to PLOS ONE. After careful consideration, we feel that it has merit but does not fully meet PLOS ONE’s publication criteria as it currently stands. Therefore, we invite you to submit a revised version of the manuscript that addresses the points raised during the review process.

**Article Report**

Article title ‘The effect of ascertainment on penetrance estimates for rare variants: implications for establishing pathogenicity and for genetic counselling’ by Andrew D Paterson et al have, developed PenEst, an app that allows users to investigate the phenomenon across ranges of parameter settings.

**Summary of article:** In this study Ascertainment effects on penetrance estimates.

*Most of the variants identified are very rare and were identified in small pedigrees, which creates challenges in terms of penetrance estimation and translation into genetic counselling in the setting of cascade testing. They illustrated robust ascertainment corrections via the LOD score, and recommend a LOD-based approach to assessing pathogenicity of rare variants in the presence of reduced penetrance.*

**Comment: Minor revision**

The work is very interesting and t*hese findings have important implications for establishing pathogenicity for variants as well as implications for cascade genetic counselling. *

*These findings will be of general interest to the human and medical genetics community since they have impact on variant interpretation and penetrance estimation for rare variants. Article must be accepted and below points may be consider for better framework of the article-*

Representation of statistical analysis should be more defined to understand better way.Results should be compared with recent published work and emphasize on betterment cause.Result and discussion section should be arranged more properly with the appropriate content.Application of the app-research defined in infectious diseases model with healthcare management.

We look forward to receiving your revised manuscript.

Kind regards,

Anshuman Mishra, PhD

Academic Editor

PLOS ONE

Journal Requirements:

Additional Editor Comments (if provided):

Dear Author,

Please find the Article Report as below and carefully submit the revised version with reviewers reply.

Article title ‘The effect of ascertainment on penetrance estimates for rare

variants: implications for establishing pathogenicity and for genetic counselling’

by Andrew D Paterson et al have, developed PenEst, an app that allows users to investigate the phenomenon across ranges of parameter settings.

Summary of article: In this study Ascertainment effects on penetrance estimates.

Most of the variants identified are very rare and were identified in small pedigrees, which creates challenges in terms of penetrance estimation and translation into genetic counselling in the setting of cascade testing. They illustrated robust ascertainment corrections via the LOD score, and recommend a LOD-based approach to assessing pathogenicity of rare variants in the presence of reduced penetrance.

Comment: Minor revision

The work is very interesting and these findings have important implications for establishing pathogenicity for variants as well as implications for cascade genetic counselling.

These findings will be of general interest to the human and medical genetics community since they have impact on variant interpretation and penetrance estimation for rare variants. Article must be accepted and below points may be consider for better framework of the article-

1. Representation of statistical analysis should be more defined to understand better way.

2. Results should be compared with recent published work and emphasize on betterment cause.

3. Result and discussion section should be arranged more properly with the appropriate content.

4. Application of the app-research defined in infectious diseases model with healthcare management.

Thanks

Regards

Dr. Anshuman Mishra

Reviewers' comments:

Reviewer's Responses to Questions

**Comments to the Author**

1. Is the manuscript technically sound, and do the data support the conclusions?

Reviewer #1: Yes

Reviewer #2: Yes

2. Has the statistical analysis been performed appropriately and rigorously? 

Reviewer #1: Yes

Reviewer #2: Yes

3. Have the authors made all data underlying the findings in their manuscript fully available?

Reviewer #1: Yes

Reviewer #2: Yes

4. Is the manuscript presented in an intelligible fashion and written in standard English?

Reviewer #1: Yes

Reviewer #2: Yes

5. Review Comments to the Author

Reviewer #1: The manuscript investigates a relevant and critical problem in the field. The study design and experimental approaches are appropriate and data supports the conclusion. Hence, I endorse publication of this manuscript.

Reviewer #2: In the current study entitled “The effect of ascertainment on penetrance estimates for rare variants: implications for 3 establishing pathogenicity and for genetic counseling”, authors performed simulation and identified that penetrance estimated for variant in rare diseases can be drastically inflated due to underlying ascertainment bias. They developed a python based tool “PenEst” for the simulation. In the end, authors recommended to use LOD-based approach to assess the pathogenicity of rare variant. The article's research is relatively clear. I endorse it for publication in PlosOne.

6. PLOS authors have the option to publish the peer review history of their article (what does this mean?). If published, this will include your full peer review and any attached files.

Reviewer #1: No

Reviewer #2: **Yes: **Manju Kashyap

---

## [Author Response · Author response to Decision Letter 0]

21 Jul 2023

The previous submission was not specifically formatted for PLOS One, so we took the advantage of the reviewer’s suggestions to better define the analysis and re-arrange the results and discussion to revise the structure of the manuscript. This has entailed including the previous supplementary text in the main manuscript, as well as major re-organization of the manuscript to make the work easier to understand. 

The reviewer’s comment about infectious disease and healthcare management are beyond the scope of the current manuscript and have not been explicitly addressed.

---

## [Editor Report · Decision Letter 1]

4 Aug 2023

The effect of ascertainment on penetrance estimates for rare variants: implications for establishing pathogenicity and for genetic counselling

PONE-D-23-06786R1

Dear Prof. Andrew D Paterson,

We’re pleased to inform you that your manuscript has been judged scientifically suitable for publication and will be formally accepted for publication once it meets all outstanding technical requirements.

Kind regards,

Anshuman Mishra, PhD

Academic Editor

PLOS ONE

Additional Editor Comments (optional):

Dear Prof. Andrew D Paterson,

Thanks for the revised article and corrections for making it more better for the readers. Hopefully this article will be helpful to understand complex genetics phenomenon through the developed PenEst for calculating and displaying the corrected penetrance estimates.

Comment: Accepted

Regards

Anshuman Mishra

PLOS ONE
---

## [Editor Report · Acceptance letter]

15 Sep 2023

PONE-D-23-06786R1 

The effect of ascertainment on penetrance estimates for rare variants: implications for establishing pathogenicity and for genetic counselling 

Dear Dr. Paterson:

I'm pleased to inform you that your manuscript has been deemed suitable for publication in PLOS ONE. Congratulations! Your manuscript is now with our production department. 

Kind regards, 

on behalf of

Dr. Anshuman Mishra 

Academic Editor

PLOS ONE